# MicroRNAs in the Pathogenesis, Prognostication and Prediction of Treatment Resistance in Soft Tissue Sarcomas

**DOI:** 10.3390/cancers15030577

**Published:** 2023-01-18

**Authors:** Andrea York Tiang Teo, Vivian Yujing Lim, Valerie Shiwen Yang

**Affiliations:** 1Yong Loo Lin School of Medicine, National University of Singapore, Singapore 117597, Singapore; 2Institute of Molecular and Cell Biology, A*STAR, Singapore 138673, Singapore; 3Division of Medical Oncology, National Cancer Centre Singapore, Singapore 169610, Singapore; 4Oncology Academic Clinical Program, Duke-NUS Medical School, Singapore 169857, Singapore

**Keywords:** soft tissue sarcomas, microRNA, prognostic biomarkers, predictive biomarkers, treatment resistance

## Abstract

**Simple Summary:**

Soft tissue sarcoma is a rare entity that accounts for 1% of adult cancers but represents 20% of paediatric solid tumours. Overall prognosis in advanced disease remains poor. MicroRNAs (miRNAs) are short non-coding RNAs that target mRNAs and control gene expression and may exert both oncogenic and tumour suppressor functions in cancers. The deregulation of miRNAs in soft tissue sarcomas may be exploited in the development of miRNA-based strategies for the prognostication of disease outcomes, identification of treatment resistance and new-generation therapeutics.

**Abstract:**

Soft tissue sarcomas are highly aggressive malignant neoplasms of mesenchymal origin, accounting for less than 1% of adult cancers, but comprising over 20% of paediatric solid tumours. In locally advanced, unresectable, or metastatic disease, outcomes from even the first line of systemic treatment are invariably poor. MicroRNAs (miRNAs), which are short non-coding RNA molecules, target and modulate multiple dysregulated target genes and/or signalling pathways within cancer cells. Accordingly, miRNAs demonstrate great promise for their utility in diagnosing, prognosticating and improving treatment for soft tissue sarcomas. This review aims to provide an updated discussion on the known roles of specific miRNAs in the pathogenesis of sarcomas, and their potential use in prognosticating outcomes and prediction of therapeutic resistance.

## 1. Introduction

Sarcomas are malignant neoplasms of mesenchymal origin with over 70 histologic subtypes and may be broadly divided into two categories: soft tissue sarcomas (thought to arise from the muscle, fat, nerve/nerve sheath, blood vessels or other connective tissue) and bone sarcomas (Figure 1) [1]. They account for 1% of adult cancers, and nearly 21% of all paediatric solid malignant cancers, with soft tissue sarcomas comprising nearly 90% of sarcomas [2]. Soft tissue sarcomas may arise anywhere in the body, but most originate in the extremities, the abdomen, or the head and neck [3,4]. While no formal etiology has yet been defined, multiple gene rearrangements have been associated with an increased risk of certain soft tissue sarcoma subtypes, such as in Ewing’s sarcoma (EWSR1–FLI-1 fusion), myxoid liposarcoma (TLS–CHOP fusion), alveolar rhabdomyosarcoma (PAX3–FHKR fusion) and synovial sarcoma (SSX–SYT fusion) [5].

Traditionally, soft tissue sarcomas are managed by wide excisional surgery for localized disease. Surgery may also be used as a palliative procedure in metastatic disease [6]. With the exception of gastrointestinal stromal tumours (GIST), adjuvant treatment is not standard, even in R_0_ resections. Radiotherapy and chemotherapy are typically reserved for advanced disease; radiotherapy is usually provided in high-risk tumours that are large, deep and/or high grade [7], while adjuvant systemic treatment is controversial, but may be considered on a case-by-case basis [8]. While a meta-analysis of randomized trials found a statistically significant—albeit marginal—advantage of adjuvant chemotherapy in terms of both recurrence-free survival and overall survival [9], a large phase III randomised controlled trial subsequently demonstrated that adjuvant chemotherapy in resected soft tissue sarcoma failed to show improved survival [10]. Increasingly, novel targeted therapies are also used in the management of soft tissue sarcomas following better understanding of the molecular pathogenesis and genomic profiles of some soft tissue sarcomas, but this applies only to a minority of subtypes [8]. The median survival in soft tissue sarcoma patients with metastatic disease is one year [11,12,13]. Even in patients with localized disease, up to 50% develop metastases and die despite undergoing definitive therapy [5,14], thus highlighting the need for earlier diagnosis, appropriate management and novel treatment approaches in soft tissue sarcomas.

**Figure 1 cancers-15-00577-f001:**
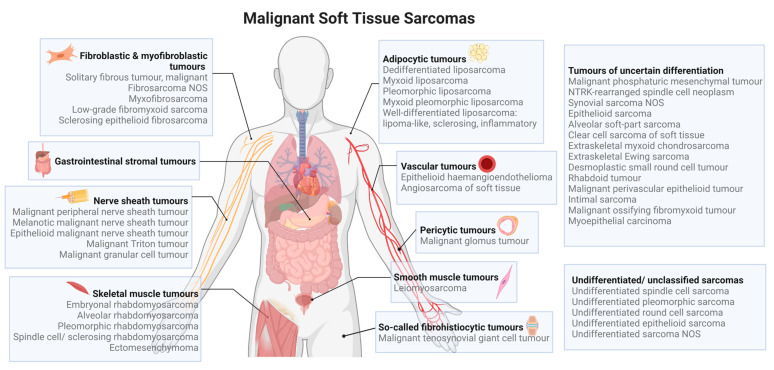
Types of soft tissue sarcomas based on the 2020 WHO classification [15].

MicroRNAs (miRNAs) are short non-coding RNAs of 19–25 nucleotides that regulate post-transcriptional gene expression. Mature miRNAs bind to complementary sites on target mRNAs, usually at the 3′ UTR, thereby suppressing mRNA translation or causing degradation of the mRNA transcript [16]. Because of their ability to target multiple different mRNAs, miRNAs are able to modulate almost any biological pathway. Accordingly, miRNAs are important regulators of various cancer-related processes, such as differentiation, proliferation, metastasis and apoptosis [17], and are therefore attractive targets for miRNA-based therapies. They have been found to be generally downregulated in tumours [18] but can exert both oncogenic and tumour suppressor functions in cancers. Although miRNAs comprise ~0.01% of the total RNA mass in a given sample, advances in strategies to detect and target miRNAs have greatly improved, thus making miRNAs attractive biomarkers in the early diagnosis, staging and monitoring of cancer progression [19], as well as targets for drug development [20].

The role of miRNAs in the diagnosis of soft tissue sarcomas has been proposed and discussed elsewhere [21,22]; however, the use of miRNA as biomarkers for predicting patient outcomes and therapeutic resistance in soft tissue sarcomas is less defined. In this review, we will first summarise the known roles of specific miRNAs in the pathogenesis of sarcomas, then discuss their potential use in prognosticating outcomes and prediction of therapeutic resistance.

## 2. MicroRNAs in the Pathogenesis of Soft Tissue Sarcomas

miRNAs mediate soft tissue sarcoma progression by influencing various pathogenic processes, thereby acting as oncogenes or tumour suppressors. As the clinical behaviours underlying pathogenesis and management of soft tissue sarcomas differ between subtypes, the following discussion will review the roles of relevant miRNAs within each soft tissue sarcoma subtype. Table 1 summarises the key miRNAs known to be involved in the regulation of various soft tissue sarcoma subtypes.

### 2.1. Gastrointestinal Stromal Tumour

Gastrointestinal stromal tumour (GIST) is the most common mesenchymal tumour specific to the gastrointestinal tract and is commonly characterized by activating mutations in the KIT or PDGFRA receptor tyrosine kinases [105]. KIT is an oncogene which has a gain-of-function mutation in approximately 70% of GISTs [106,107]. Several miRNAs are shown to inhibit the progression of GIST via the regulation of KIT (Figure 2).

Downregulation of miR-494 has been observed in GIST cell lines, and miR-494 overexpression in GIST cells triggered apoptosis and inhibited cell growth [23,24]. miR-494 was found to regulate the expression of KIT and other molecules in its downstream signaling cascade, including phospho-AKT and phospho-STAT3 [23]. miR-494 was also shown to target survivin, with downregulation of survivin leading to G2-M phase arrest and apoptosis, along with inhibition of cell proliferation and colony formation [24]. Analysis of survivin/KIT interaction showed that survivin regulated KIT expression at the transcription level, thus exerting effects on the PI3K-AKT pathway in GIST as well [24]. Another miRNA found to be markedly decreased in GIST tissues is miR-218 [25,26], with ectopic overexpression in GIST cells via chitosan-tocopherol nanoparticle or liposome delivery demonstrating decreased cell proliferation and increased apoptosis [25,27]. KIT was identified as a target of miR-218 in both studies [25,27]. The miR-221/222 cluster, dysregulated in many malignancies [108,109,110], is also downregulated in GIST [28,29,30]. KIT-positive GISTs showed significant repression of miR-221/222 as compared to normal tissues and KIT-negative GISTs [28]. The role of miR-221/222 in the modulation of KIT and the PI3K/AKT pathway in GIST was confirmed by Ihle et al. who demonstrated that transient transfection of miR-221/222 reduced GIST cell viability and induced apoptosis by inhibition of KIT expression and its downstream signalling cascade [29]. This was corroborated by Gits et al. who showed the direct regulation of KIT by miR-222 in GIST [30].

Other miRNAs that are downregulated in GIST include miR-17, miR-20a, miR-4510, miR-133b, miR-152, miR-518-5p and miR-137 [30,31,32,33,34]. These miRNAs play a role in controlling tumour proliferation, migration and invasion, and in inducing apoptosis. miR-17 and miR-20a act by targeting ETV1 [30], a transcription factor that supports tumorigenesis and is universally highly expressed in GISTs [111]. miR-4510 demonstrated its tumour suppressor effects by targeting and inhibiting apolipoprotein C-II (ApoC2) expression, and also decreased the activity of AKT, ERK1/2, MMP2 and MMP9 [31]. miR-152 was found to target and suppress the expression of cathepsin L (CTSL) [32], a lysosomal cysteine protease correlated with metastatic aggressiveness and poor patient prognosis [112]. Epithelial mesenchymal transition (EMT) was another key process in cancer progression regulated by miR-137, which was reported to target Twist1 and increase the expression of epithelial markers E-cadherin and cytokeratin while decreasing the expression of mesenchymal markers N-cadherin and vimentin in GIST cells. As with the other aforementioned miRNAs, miR-137 also decreased GIST cell migration, activated G1 cell cycle arrest, and induced cell apoptosis [35].

On the contrary, oncogenic miRNAs promote the development of GIST. miR-374b was highly expressed in GIST tissues, and its expression increased the mRNA and protein levels of various molecules in the PI3K/AKT cell survival pathway in GIST cells [36]. miR-374b was also found to promote cell viability, migration, invasion and cell cycle progression in GIST cells, along with inhibition of apoptosis [36]. It was further reported that miR-196a expression is overexpressed in high-risk GIST samples as compared to the low- or intermediate-risk GIST tissues, with the upregulation of miR-196a associated with GIST malignancy [37].

### 2.2. Liposarcoma

Liposarcoma is one of the most common soft tissue sarcomas, and may be classified into four subtypes based on its pathological and molecular genetic characteristics: pleomorphic (PLPS), myxoid/round cell (MLPS/RLPS), dedifferentiated (DDLPS) and well-differentiated (WDLPS). The round cell component in MLPS/RLPS is thought to be associated with metastasis and poorer prognosis [113]. The regulation of liposarcoma by miRNAs occurs through various mechanisms (Figure 3).

miR-155 is known to act as an oncogene in multiple malignant tumours [114]. It has been found to be the most over-expressed miRNA identified in DDLPS tumour samples and cell lines [46] and is also over-expressed in PLPS and MLPS/RLPS [39,47]. miR-155 was further shown to promote tumour cell growth in DDLPS by targeting casein kinase 1α (CK1α), thereby enhancing β-catenin signalling and cyclin D1 expression [46]. miR-26a-2 has also been found to be overexpressed in WDLPS, DDLPS and MLPS/RLPS [38,49]. Overexpression of miR-26a-2 in LPS cell lines improved sarcoma cell growth and survival, including faster cell proliferation and migration, enhanced clonogenicity, suppressed adipocyte differentiation and/or resistance to apoptosis. Overexpression of RCBTB1, a direct target of miR-26a-2, made LPS cells more susceptible to apoptosis [49]. HOXA5 has also been demonstrated to be a target of miR-26a-2, with the downregulation of HOXA5 inhibiting the apoptotic response in LPS cells [50].

Other miRNAs which are overexpressed in LPS and play a role in invasion and metastasis include miR-135b, miR-25-3p and miR-92a-3p [51,52,53]. miR-135b is highly expressed in the round cell component of MLPS/RLPS and has been found to promote MLPS/RLPS cell invasion in vitro and metastasis in vivo by targeting the expression of thrombospondin 2 (THBS2). Decreased THBS2 expression increases the amount of matrix metalloproteinase 2 (MMP2) thereby modulating the extracellular matrix structure, resulting in a morphological change of the tumour [51]. A study on the extracellular vesicles secreted by LPS cells showed that they contained miR-25-3p and miR-92a-3p, though they were downregulated within the liposarcoma tumour itself. The secretion of miR-25-3p and miR-92a-3p then initiated the release of proinflammatory cytokine IL-6 from tumour-associated macrophages, in turn enhancing LPS cell proliferation, invasion and metastasis [52]. More recently, miR-1246, miR-4532, miR-4454, miR-619-5p and miR-6126 have been identified to be highly expressed in human DDLPS cell lines and exosomes and are believed to promote tumour progression [115].

On the contrary, certain miRNAs have been found to inhibit the progression of LPS through suppression of proliferation and induction of apoptosis. miR-143 is downregulated in both WDLPS and DDLPS tumours and cell lines [38,39]. Restoring miR-143 expression in DDLPS cells decreased the expression of BCL2, topoisomerase 2A, protein regulator of cytokinesis 1 (PRC1), and polo-like kinase 1 (PLK1). It was further shown that treatment of LPS cells with a PLK1 inhibitor potently arrested cytokinesis in the G2–M phase and induced apoptosis [38]. miR-486 expression was also found to be repressed in MLPS tissues, with the restoration of miR-486 expression resulting in repressed MLPS cell growth [40]. Plasminogen activator inhibitor-1 (PAI-1), shown to promote tumour invasion and angiogenesis [116], was identified as a target of miR-486. Accordingly, knockdown of PAI-1 inhibited the growth of MLPS cells [40]. Another miRNA that inhibits LPS cell growth and migration in vitro and suppresses tumour growth in vivo is miR-195. Oxysterol-binding protein (OSBP) was demonstrated as a direct target of miR-195, with the overexpression of OSBP reversing the effects of miR-195 on LPS cell growth, migration and apoptosis [45]. miR-145 and miR-451 expression have also been found to be reduced in human LPS samples of all subtypes [38,39,41], with the reintroduction of miR-145 and miR-451 in LPS cell lines resulting in impaired cell cycle progression and cellular proliferation and increased cellular apoptosis [41].

Some miRNAs may also regulate LPS progression by interfering with tumour cell metabolism and introducing oxidative stress. miR-133a, significantly underexpressed in DDLPS tissues, has been found to modulate DDLPS cell metabolism, with enforced expression of miR-133a resulting in decreased glycolysis and increased oxidative phosphorylation. This was coupled with impaired cell proliferation and cell cycle progression [44]. miR-193b is found to be underexpressed in DDLPS, with exogenous reintroduction of miR-193b resulting in LPS cell apoptosis [42]. miR-193b targets CRK-like proto-oncogene (CRKL) and focal adhesion kinase (FAK); in vivo studies to introduce miR-193b mimetics and an FAK inhibitor resulted in inhibited LPS xenograft growth in both cases. In addition, miR-193b also induced oxidative stress in LPS cells by targeting an antioxidant, methionine sulfoxide reductase A (MsrA) [42]. Further studies revealed that miR-193b also directly targets PDGFRβ, SMAD4, and YAP1 [43]. Inhibition of PDGFRβ attenuates the differentiation and proliferation of LPS cells, while knockdown of SMAD4 promotes adipogenic differentiation. Direct inhibition of YAP1 reduces the activity of Wnt/β-catenin signalling. Subsequent introduction of a PDGFR inhibitor and a Wnt/β-catenin inhibitor demonstrated reduced cell viability and increased apoptosis in DDLPS and WDLPS cells [43]. 

### 2.3. Rhabdomyosarcoma

Rhabdomyosarcoma (RMS) is the most common soft tissue sarcoma in paediatric patients and young adults. RMS can be classified into two major histological subtypes: embryonal rhabdomyosarcoma (ERMS) and alveolar rhabdomyosarcoma (ARMS). RMS may be further classified based on clinical outcome into fusion-positive RMS or fusion-negative RMS based on the presence or absence of either PAX3-FOXO1 or PAX7-FOXO1 gene fusions. As these gene fusions are absent in ERMS, ERMS patients are all fusion-negative, while majority of ARMS patients are fusion positive [117,118,119]. Fusion-positive RMS tends to have a worse prognosis and overall survival than fusion-negative RMS, thus ARMS is associated with poorer prognosis [120,121,122].

In recent studies, miRNAs which have been identified to play a role in skeletal muscle proliferation and differentiation such as miR-1, miR-133, miR-206 and miR-29 [123,124], have been investigated for their roles in RMS. miR-206 plays an important role in the regulation of RMS, with multiple studies demonstrating downregulation of miR-206 in RMS tissues and cell lines as compared to human myotubes and skeletal muscle [54,55,56,57,58,59,60,61]. Exogenously increasing miR-206 levels in RMS has been shown to promote myogenic differentiation and block tumour growth in xenografted mice by switching the global mRNA expression profile to one that resembles mature muscle [54]. This was corroborated by a separate study showing that the activation of miR-206 resulted in a genetic switch in RMS cells from a proliferative growth phase to differentiation [57]. In ERMS, the following regulation pathway of miR-206 was uncovered: PAX3/7-FOXO1 induced oxidative stress response factor HO-1 expression, which in turn resulted in miR-206 repression. HO-1 inhibition showed reduced RMS tumour growth and vascularisation in vivo, accompanied by the induction of miR-206 [60]. miR-206 then exerts its anti-tumorigenic effects by targeting and suppressing the Met receptor tyrosine kinase (c-Met), which is overexpressed in both ARMS and ERMS [125], and has been implicated in RMS pathogenesis [54,55]. SMYD1 silencing, which occurs with low levels of miR-206 in RMS, impairs differentiation of all subtypes of RMS. On the contrary, silencing of G6PD, a direct target of miR-206, successfully suppressed RMS cell proliferation and growth [59]. Furthermore, ectopic expression of miR-206 in a ERMS fusion-negative RMS cell line showed significant downregulation of PAX3 protein expression, but this was not observed in ARMS fusion-positive RMS cells as the formation of a fusion transcript between PAX3 and FOXO1 enabled the cells to evade miRNA-mediated regulation of PAX3 [56]. In addition, PAX7 downregulation was shown to be essential for miR-206-induced cell cycle exit and myogenic differentiation in fusion-negative RMS but not in fusion-positive RMS. Genetic deletion of miR-206 in a mouse model of fusion-negative RMS promoted tumor development [58]. Interestingly, while there is much evidence to show that miR-206 is downregulated in RMS tumours and cell lines, analysis of plasma samples of RMS patients has found significantly increased levels of miR-206 as compared to healthy individuals and patients with non-RMS tumours [67,126]. This may be because RMS forms within skeletal muscle, and miR-206 is a muscle-specific miRNA, thus elevated levels of this miRNA may be found in RMS patient serum.

Another notable miRNA in RMS regulation is miR-1 [54,55,63]. Besides downregulating PAX3 expression in ERMS [56], miR-1 was also found to suppress c-Met expression in RMS [55]. miR-1 was shown to encourage myogenic differentiation in RMS cells, and ectopic increase in miR-1 expression resulted in growth inhibition of RMS cells, likely due to G1-S cell cycle arrest [62]. Furthermore, overexpression of miR-1 and miR-133b also resulted in autophagic cell death through the silencing of polypyrimidine tract-binding protein 1 (PTBP1), a positive regulator of cancer-specific energy metabolism [63]. PAX3-FOXO1, which could upregulate the expression of a key kinase involved in glycolysis and the Warburg effect through increased expression of PTBP1, was targeted and repressed by miR-133b [63]. Overexpression of miR-133a in ERMS cells resulted in cell cycle arrest, suggesting its role as a tumour suppressor [62].

Several miRNAs share similar targets in the regulation of RMS. miR-29 is a key miRNA that is epigenetically silenced in RMS tissues and cell lines [56,64,65]. It has been reported that the downregulation of miR-29 occurs via an activated NF-κB-YY1 pathway, in which NF-κB acts through Ying Yang 1 (YY1) [64]. miR-29 was found to target and repress the expression of cell cycle regulators cyclin D2 and E2F7, resulting in partial G1 arrest and decreased cell proliferation in RMS [56]. In addition, miR-29 also targets GEFT, which is associated with poor prognosis in patients with RMS [127]. Repression of GEFT activity by miR-29 weakened the effect of GEFT on the migration, invasion and apoptosis of RMS cells [65] while restoration of miR-29 in mice inhibited tumour growth and stimulated differentiation [64,65]. GEFT translation and expression were also found to be inhibited by miR-874, an miRNA downregulated in RMS tissues. Overexpression of miR-874 in RMS cells inhibited proliferation, invasion and migration in RMS cells and also induced apoptosis, while GEFT restoration partially reversed the anti-tumour effects of miR-874 [77]. miR-26a has also been found to be downregulated in RMS tumours and cell lines. Expression levels of miR-26a were further demonstrated to be inversely related to EZH2 [66], a histone methyltransferase overexpressed in various aggressive cancers [128,129]. Circulating levels of miR-26a in RMS patient plasma were also reduced, and miR-26a plasma levels were associated with fusion status, with PAX3/7-FOXO1-positive RMS samples displaying lower levels of miR-26a compared to fusion-negative samples [67]. Separately, EZH2 was found to downregulate miR-101 in ERMS cells via a negative feedback loop, and overexpression of miR-101 was able to reduce ERMS tumorigenic potential, impairing colony formation and cell cycle progression [76]. 

Other miRNAs which are downregulated in RMS cell lines and tissue samples have also demonstrated anti-tumour effects in RMS by inducing apoptosis and myogenic differentiation, as well as impairing cell proliferation, invasion and metastasis. In miR-7 transfected RMS cells, miR-7 acts via its target mitochondrial proteins solute carrier family 25 member 37 (SLC25A37) and translocase of inner mitochondrial membrane 50 (TIMM50) to promote apoptosis and necroptosis [68], and also impair tumour invasion and lung metastasis [69]. In addition, miR-7 and miR-324-5p regulate pro-oncogenic protein ITGA9, and overexpression of the two miRNAs reduced tumour growth in orthotopic mice tumour models [69]. Insulin-like growth factor receptor 1 (IGF1R), a key signalling molecule in RMS, was shown to be a target of miR-378a-3p. Upregulation of miR-378a-5p expression resulted in apoptosis, decreased cell viability and G2 phase cell cycle arrest in RMS cells, along with upregulation of myogenic proteins such as MyoD and MyHC, demonstrating a shift towards myogenic differentiation [70]. Similarly, miR-450b-5p, suppressed in RMS by TGF-β1 through a pathway mediated by Smad3 and Smad4, exerted anti-tumour effects in tumour implants and cells by arresting RMS growth and upregulating MyoD expression [71]. An autoregulatory loop between TGF-β1/miR-411-5p/SPRY4 and MAPK in RMS has also been established, in which it is suggested that miR-411-5p inhibits SPRY4 to activate MAPK, promoting apoptosis and myogenic differentiation in RMS cells [73]. miR-203 was also reported to be downregulated by promoter hypermethylation in RMS tumour samples and cell lines. Restoration of miR-203 expression in RMS cells was able to inhibit their migration and proliferation. Furthermore, miR-203 promoted myogenic differentiation by inhibiting the Notch and JAK1/STAT1/STAT3 pathways via its target proteins p63 and leukaemia inhibitory factor receptor [72]. miR-214 reintroduction into RMS cells was able to inhibit tumour cell growth, promote apoptosis and induce myogenic differentiation. Proto-oncogene N-ras was reported as a conserved target of miR-214 in its suppression of xenograft tumour growth [75]. Lastly, exogenous expression of miR-410-3p was shown to inhibit EMT in RMS, with the inhibition of RMS cell invasion, migration and proliferation [78]. 

In contrast, miR-27a and miR-486-5p were discovered to be upregulated in the more aggressive fusion-positive RMS samples and cell lines [74,79]. miR-27a was further found to enhance cell cycle progression by targeting the retinoic acid alpha receptor (RARA) and retinoic X receptor alpha (RXRA), resulting in increased RMS cell proliferation [79]. Regulation of miR-27a via a HDAC3–SMARCA4–miR-27a–PAX3-FOXO1 circuit further demonstrated the ability of miR-27a to destabilize PAX3-FOXO1 mRNA in ARMS cells [80].

### 2.4. Malignant Peripheral Nerve Sheath Tumour

Malignant peripheral nerve sheath tumour (MPNST) is an aggressive soft tissue sarcoma arising from peripheral nerves or deep neurofibromas. It has a poor prognosis due to its propensity for metastasis and local recurrence. MPNSTs may occur sporadically, but around half of MPNST cases arise in patients with the autosomal dominant genetic disorder neurofibromatosis type 1 (NF1) [130,131]. Most NF1-related tumours demonstrate abnormal Ras signalling pathways, with various genes involved in the Ras pathway deregulated in MPNSTs [132].

miR-204 was found to be downregulated in both NF1 and non-NF1 MPNST tissues and cell lines [81]. Restoration of miR-204 levels resulted in reduced cellular proliferation, migration and invasion in vitro, and decreased tumour growth and invasion in vivo. It was further found that miR-204 modulated Ras signalling and carcinogenesis progression in MPNSTs via direct inhibition of HMGA2 [81]. Members of the miR-30 family have also been reported to be downregulated in MPNSTs. The transcription of miR-30d is inhibited by high levels of enhancer of zeste homolog 2 (EZH2) in MPNST, thereby leading to enhanced expression of karyopherin beta 1 (KPNB1), a direct target of miR-30d. Exogenous regulation of the EZH2–miR-30d–KPNB1 signalling pathway was able to induce MPNST cell apoptosis in vitro and suppress tumorigenesis in vivo [82]. A further study showed that miR-30a demonstrated similar regulation of expression in MPNSTs via the EZH2–miR-30a–KPNB1 signalling pathway [83]. MiR-200b was found to be suppressed by EZH2, resulting in EMT in MPNST cells, often thought to be one of the initial steps in metastasis [83]. Finally, miR-34a expression is downregulated in MPNSTs relative to neurofibromas due to p53 inactivation, with exogenously increased expression of miR-34a demonstrating increased apoptotic cell death [84]. 

Likewise, certain miRNAs in MPNSTs are upregulated. miR-21 expression levels in MPNST clinical samples were significantly higher compared to NF samples [85], congruent to its high levels of expression in multiple other types of cancers and soft tissue tumours [133]. Inhibition of miR-21 in MPNST cell lines showed suppressed cell growth and upregulated levels of its target protein, programmed cell death protein 4 (PDCD4), which is known to act as a tumour suppressor gene and is upregulated during apoptosis [134]. It was further found that miR-21 inhibition decreased caspase activity, suggesting that miR-21 plays a crucial role in modulating programmed cell death in MPNSTs [85]. 

### 2.5. Leiomyosarcoma

Leiomyosarcomas (LMS) are highly aggressive malignancies of smooth muscle tissues which account for approximately 10% of all STS [135]. Uterine leiomyosarcomas (UMLS) account for the single largest site-specific group of LMS and is also the most common subtype of uterine sarcomas [135,136].

There are statistically significant differences in the expression of multiple miRNAs between LMS and smooth muscle samples [137], endometrial stromal sarcomas [138] and undifferentiated pleomorphic sarcomas [91], pointing towards a unique miRNA signature which could be used for the detection and diagnosis of LMS. In LMS, high levels of miR-181b were observed in both ULMS and soft tissue LMS, though to a greater extent in ULMS [90]. On the contrary, miR-152 down regulation was observed in LMS samples [89]. Transfection of miR-152 into LMS cells resulted in decreased proliferation, increased apoptosis and S-phase cell cycle arrest. This was coupled with downregulation of proto-oncogenes MET and KIT mRNA and protein expression, which in turn was associated with a transient down-regulation of the PI3K/AKT pathway [89]. In addition, overexpression of maternal embryonic leucine zipper kinase (MELK), an oncogenic kinase [139,140], in ULMS showed significant downregulation of miR-34a expression. The IL-6 receptor was identified as the target gene of miR-34a, such that decreased miR-34a could induce the activation of the JAK2/STAT3 pathway and a consequent anti-apoptotic mechanism [88]. 

### 2.6. Synovial Sarcoma

Synovial sarcoma is a high-grade mesenchymal neoplasm that accounts for 10% to 20% of all soft tissue sarcomas in adolescents and young adults [141,142]. 

Downregulation of miR-494-3p and miR-126 was discovered in synovial sarcoma tumours [92,93]. Re-expression of miR-494-3p in synovial sarcoma cells was associated with a decrease in cell proliferation and migration, along with apoptosis induction. CXCR4, involved in tumour development and metastatic spread in a variety of cancers [143,144], as well as synovial sarcoma cell migration and invasion [145], has been identified as a potential target of miR-494-3p [92]. The long non-coding RNA HOTAIR was shown to regulate the expression of miR-126 in synovial sarcoma, and miR-126 in turn targeted SDF-1, a protein that modulates EMT, migration and proliferation in synovial sarcoma [93].

Likewise, oncogenic miRNAs involved in the potentiation of synovial sarcoma were also reported. Overexpression of let-7e microRNA and miR-99b in synovial sarcoma were found, and the downregulation of the two miRNAs using miRNA inhibitors resulted in the suppression of cell proliferation, accompanied by an increased expression of their putative targets, high mobility group (HMGA2) and SMARCA5 [94], both of which are associated with the development of tumours [146,147,148,149]. miR-214 also played a role in synovial sarcoma development by enhancing cytokine expression, though there was no evidence to suggest that it could induce cellular growth, migration or invasion [96]. In addition, miR-9 was found to induce EMT in synovial sarcoma cell via its target protein CDH1, thereby activating associated MAPK/ERK and Wnt/β-catenin signalling pathways and eliciting pro-tumorigenic effects and inhibiting apoptosis [97]. miR-17, expressed and upregulated in synovial sarcoma, was shown to target p21 [98], a tumour suppressor shown to induce growth arrest and differentiation in cancers [150]. Knockdown of miR-17 in turn showed significantly decreased cell growth [98]. 

### 2.7. Fibrosarcoma

Fibrosarcomas are defined as a malignant neoplasm composed of fibroblasts with variable collagen production [151]. miR-197-3p is downregulated in human fibrosarcoma cells [100]; restoration of miR-197-3p levels inhibits fibrosarcoma cell viability, colony forming and migration ability, and triggers G2-M phase cell cycle arrest and autophagy. Ras-related nuclear protein (RAN) which is overexpressed in various cancers [152,153,154], has been identified as a direct target of miR-197-3p. Exogenous expression of miR-197-3p resulted in the suppression of RAN and the consequent attenuation of fibrosarcoma cell proliferation and migration [100]. The miR-29 family (miR-29s) has also been found to be under expressed in human fibrosarcoma cells [99]. It was further discovered that MMP2, a pro-tumorigenic pro-angiogenic enzyme commonly overexpressed in metastatic cancer [155,156], is a direct target of miR-29s. Ectopic expression of miR-29s resulted in reduced MMP2 enzyme activity and inhibition of fibrosarcoma cell invasion [99]. Conversely, miR-520c and miR-373, overexpressed in fibrosarcoma cells, directly target mTOR and SIRT1, which are negative regulators of MMP9 expression. An ectopic increase in miR-520c and miR-373 levels therefore demonstrated a resultant increase in MMP9 activity, enhancing cell migration and growth [101].

### 2.8. Angiosarcoma

Angiosarcomas are vascular sarcomas of endothelial cell origin [157]. Three microRNAs, miR-497-5p, miR-210 and miR-340 act as tumour suppressors and are downregulated in angiosarcomas [102,103,104]. Theintroduction of miR-497-5p mimics in vitro inhibited cell proliferation, cell cycle progression, and invasion by downregulating MMP9 and cell cycle related proteins cyclin D1 and p53. miR-497-5p was also found to target and repress the calcium-activated potassium channel KCa3.1, such that the use of a KCa3.1 inhibitor or miR-497-5p mimics in an in vivo angiosarcoma xenograft inhibited tumour growth [102]. miR-210 was shown to target E2F transcription factor 3 and ephrin A3, in which knockdown of the two proteins resulted in angiosarcoma cell number reduction [103]. Finally, overexpression of miR-340, an established tumour-suppressor in multiple cancers [158,159,160], demonstrated growth inhibition and reduced invasion in angiosarcoma cells. Sirtuin 7 (SIRT7) was identified as a target gene of miR-340, with silencing of SIRT7 resulting in the inhibition of angiosarcoma cell proliferation and invasion [104].

## 3. MicroRNAs in Prognostication of Soft Tissue Sarcomas

Recent studies have also reported the correlations between miRNA expression and metastatic risk, tumour grade, overall survival and recurrence-free survival, indicating the possible utility of miRNA-guided prognostication of soft tissue sarcomas. A summary of the miRNAs involved in the prognostication of soft tissue sarcomas is found in Table 2.

### 3.1. Gastrointestinal Stromal Tumour

In GIST, smaller tumour size and a lower mitotic rate correspond with lower metastatic risk [165]. Negative correlations between miR-494 expression and tumour size, mitotic index, grade and survival were found. Kaplan–Meier analysis revealed that patients expressing weak levels of miR-494 had poorer overall survival [24]. miR-133b, downregulated in GIST, has been found to target and suppress the expression of fascin-1 directly. Elevated levels of fascin-1 expression were significantly correlated with shorter disease-free survival, and several pathological features associated with a more aggressive phenotype and metastasis, such as tumour size, mitotic counts, risk grade, blood vessel invasion and mucosal ulceration [33]. In addition, low expression of miR-1915 has been correlated with metastasis, shorter disease-free survival and overall survival using Kaplan-Meier analysis [161]. Low miR-186 levels in GIST are also associated with metastatic recurrence and a poor prognosis, with the inhibition of miR-186 resulting in the upregulation of a set of genes implicated in cancer metastasis [163]. Furthermore, under-expression of let-7e miRNA and the overexpression of its target genes were associated with poorer relapse-free survival [162]. miR-215-5p expression and the risk grade of GIST were also negatively correlated [164]. In contrast, overexpression of miR-196a in GIST was associated with a high-risk grade, a greater propensity for metastasis and poor survival [37].

### 3.2. Liposarcoma

Kaplan–Meier survival analysis revealed that overexpression of miR-26a-2 was significantly correlated with poor patient survival in WDLPS, DDLPS and MLPS [49]. The expression levels of miR-135b and THBS2 were associated with a higher risk of metastasis, and accordingly correlated significantly with a poorer prognosis in MLPS/RLPS patients [51]. Furthermore, miR-155 has been found as an indicator of unfavourable prognosis in LPS, with higher miR-155 expression levels associated with a worse overall survival rate and relapse-free survival [39].

### 3.3. Rhabdomyosarcoma

In RMS, certain miRNAs are differentially expressed between fusion-negative RMS and fusion-positive RMS (RMS with either PAX3-FOXO1 or PAX7-FOXO1 fusion oncogenes) and therefore point toward varied clinical outcomes. Low miR-206 expression was correlated with poor overall survival and was an independent predictor of shorter survival in metastatic ERMS and fusion-negative ARMS. Low miR-206 expression also significantly correlated with high SIOP stage and the presence of metastases at diagnosis [61]. Lower levels of circulating miR-26a were found to be present in patients with fusion-positive RMS as compared to fusion-negative RMS, and patients with progressive disease and poorer overall and progression-free survival showed lower levels of miR-26a as well [67]. Furthermore, the PAX3-FOXO1 fusion protein, present in fusion-positive RMS, repressed miR-221/222 that exerts anti-tumorigenic effects on RMS through the negative regulation of cyclin D2, CDK6 and ERBB3. In contrast, PAX3-FOXO1 transcriptionally upregulates miR-486-5p expression and promotes fusion-positive RMS proliferation, invasion and colony formation [74]. 

### 3.4. Leiomyosarcoma

In LMS, miR-181b-5p was associated with recurrence-free survival, and high miR-181b levels were found to be an independent predictor of recurrence-free survival regardless of LMS subtype and tumour size [90]. Furthermore, a comparison between primary and metastatic ULMS lesions showed relative overexpression of miR-15a and miR-92a in metastatic ULMS, while miR-31 was relatively overexpressed in primary lesions instead [138]. These three miRNAs control the expression of six different genes that are part of the Wnt signalling pathway, including the Frizzled-6 precursor (FZD6) gene, which was found to be of higher levels in metastatic ULMS samples. Subsequent siRNA silencing of Frizzled-6 inhibited cellular invasion and impaired MMP2 activity in ULMS cells [138].

### 3.5. Synovial Sarcoma

In metastatic tumour samples of synovial sarcoma, downregulation of miR-494-3p and increased expression of its potential target CXCR4 were more pronounced than in non-metastatic tumours and healthy tissues [92]. High expression levels of miR-214 in synovial sarcoma tumours were also correlated with poor prognosis and shorter overall survival [96]. Interestingly, the correlation between serum miR-92b-3p levels and tumour size was observed to be statistically significant, thus suggesting that serum miR-92b-3p levels could reflect tumour burden in synovial sarcoma patients [95].

## 4. MicroRNAs in Treatment Resistance

miRNAs have also been shown to play a role in influencing the resistance of soft tissue sarcomas to various forms of treatment (Figure 4). Reversal of miRNA expression in resistant tumours has the ability to modulate tumour progression, demonstrating the potential utility of miRNA-based therapy in the treatment of soft tissue sarcomas.

Imatinib, a tyrosine kinase inhibitor, works by inhibiting KIT activation, thereby blocking the activation of the downstream MAP kinase and PI3K-AKT cell survival pathways. Due to the prevalence of GISTs harbouring activating mutations in KIT, clinical management of metastatic or recurrent GIST usually involves the use of a tyrosine kinase inhibitor such as imatinib mesylate [166]. However, a number of GISTs may progress during or after treatment with imatinib, posing a challenge to clinicians [167]. Recent studies show that miRNAs play a role in modulating imatinib resistance in GIST, which could be useful in guiding clinical management of imatinib-resistant GISTs, or even using miRNAs as a novel therapeutic tool in the treatment of GIST. 

miR-218 expression was found to be decreased in imatinib-resistant cell lines, but subsequent ectopic overexpression of miR-218 in imatinib-resistant cells under the effect of imatinib mesylate resulted in significantly decreased cell viability and increased apoptosis. It was further suggested that the PI3K/AKT signalling pathways could play a role in this mechanism [26]. Similarly, miR-21 increased the susceptibility of GIST cells to imatinib, with miR-21-transfected GIST cells demonstrating increased growth inhibition and apoptosis in response to imatinib treatment compared to controls [168]. miR-518a-5p, downregulated in imatinib-resistant GISTs, was able to reduce imatinib-resistant GIST cell proliferation and increase apoptosis when introduced exogenously. It is suggested that modulation of PIK3C2A, the direct target of miR-518a-5p, affects the cellular response of GIST to imatinib mesylate, thereby causing resistance [34]. Lower levels of miR-30a were also detected in GIST cells with lower sensitivity to imatinib treatment, with imatinib treatment further reducing miR-30a levels in GIST cells. miR-30a was found to increase susceptibility to imatinib via Beclin-1 knockdown, which increased imatinib sensitivity in GIST cells. These results were confirmed in mouse tumour models [169]. miR-130a suppression by the long non-coding RNA HOTAIR increased autophagy and promoted imatinib-resistance in GIST, through its target autophagy-related protein 2 homolog B (ATG2B) [170]. Interestingly, GISTs with lower miR-320a expression showed significantly shorter time to imatinib resistance, though the mechanism through which this was mediated was not indicated in the study [171].

In contrast, overexpression of miR-125a-5p and miR-107 was associated with imatinib resistance in GIST specimens. It was further shown that expression of the miR-125a-5p target PTPN18 was suppressed in imatinib-resistant GIST samples, and that the silencing of PTPN18 expression increased cell viability in GIST882 cells with a homozygous KIT mutation subsequent to imatinib treatment. However, miR-125a-5p expression did not modulate imatinib sensitivity in GIST48 cells with double KIT mutations [161]. It was also observed that higher expression levels of phosphorylated FAK (pFAK), a downstream target of PTPN18, were present in a GIST cell line with acquired imatinib resistance as compared to its imatinib-sensitive parental cells. High FAK and pFAK levels were also associated with KIT mutation status in clinical GIST samples. Treatment with a FAK inhibitor showed that it could reverse the imatinib-resistance effect due to miR-125a-5p overexpression and cause reduced cell viability and increased apoptosis with imatinib treatment [172].

The use of miRNAs as potential biomarkers of imatinib resistance in GIST was studied by Kou et al. who found that serum miR-518e-5p could discriminate imatinib-resistant GIST patients from healthy controls and imatinib-sensitive GIST patients [173]. This could have potential implications in the way detection and diagnosis of imatinib resistance is made, thereby influencing clinical management. 

In ARMS, the PAX3-FOXO1 fusion oncogene regulates chemotherapy and radiotherapy tolerance [120]. Repression of the oncogenic miR-27a was found to play a role in PAX3-FOXO1 mRNA destabilization and increased susceptibility of RMS models to the chemotherapy drug vincristine. This downregulation of miR-27a could be achieved through the use of the histone deacetylase inhibitor entinostat, which repressed the activity of the chromatin remodeling enzyme SMARCA4 by inhibiting HDAC3 expression, thereby downregulating miR-27a [80].

Higher levels of MELK were associated with doxorubicin chemoresistance in ULMS cells. MELK overexpression in ULMS could induce M2 macrophage polarization via the miR-34a/JAK2/STAT3 pathway, contributing to doxorubicin chemoresistance in the tumour microenvironment [88]. In synovial sarcoma, miR-17 was able to confer doxorubicin resistance by reversing the effects of doxorubicin in p21 expression [98]. 

A summary of the miRNAs implicated in treatment resistance in soft tissue sarcomas may be found in Table 3.

## 5. Conclusions and Future Directions

The exciting field of miRNA research has seen miRNA-based technology entering pre-clinical and clinical settings as diagnostic and therapeutic tools for various diseases in recent years. At present, several miRNA-targeted therapeutics for cancer have reached clinical development, including the use of an miR-34 mimic (MRX34) encapsulated in lipid nanoparticles for the treatment of multiple solid tumours and an miR-16 mimic for the treatment of lung cancer [174]. However, important challenges to the clinical use of miRNA-based therapies remain. The ability of miRNAs to target multiple different mRNAs is a double-edged sword—while a single miRNA can regulate multiple cancer-related pathways, off-target effects in healthy cells remain a significant concern. Thus, identification of miRNAs specific to cancer cells, and directed delivery of miRNAs to target sites to eliminate this risk is crucial. Furthermore, the delivery of miRNAs is in itself one of the biggest hurdles for miRNA advancement into the clinical setting. Delivery associated toxicity, immune response, and difficulties in transfection and biodistribution are but some of the barriers facing safe and efficient miRNA delivery [175]. Therefore, the need for rigorous evaluation of toxicity and target engagement is required to avoid early failure in clinical trials. 

In this review, three key areas in which miRNAs may be utilised in the management of soft tissue sarcomas have been discussed: (i) prognostication, (ii) prediction of treatment resistance, and (iii) therapeutics. The use of miRNAs in the prognostication of soft tissue sarcomas could guide clinical management by identifying patients who have higher metastatic risk and thus require closer surveillance or a lower threshold for adjuvant treatment. Studying miRNA expression in patient serum could also serve as a biomarker for the earlier detection of disease relapse. Distinguishing which cancers are more amenable to treatment options based on their miRNA signature could also increase the overall efficacy of soft tissue sarcoma therapy. Furthermore, the prediction of treatment resistance to specific agents can potentially guide systemic treatment choices in advanced soft tissue sarcomas. Finally, miRNA-based therapies offer an appealing approach to cancer treatment because a single miRNA can regulate multiple target genes and/or signalling pathways within cancer cells. While few miRNA-based strategies have reached clinical routine, continuous advancements in this field offer great promise for their utility in diagnosing, prognosticating and improving treatment outcomes for soft tissue sarcomas.

## Figures and Tables

**Figure 2 cancers-15-00577-f002:**
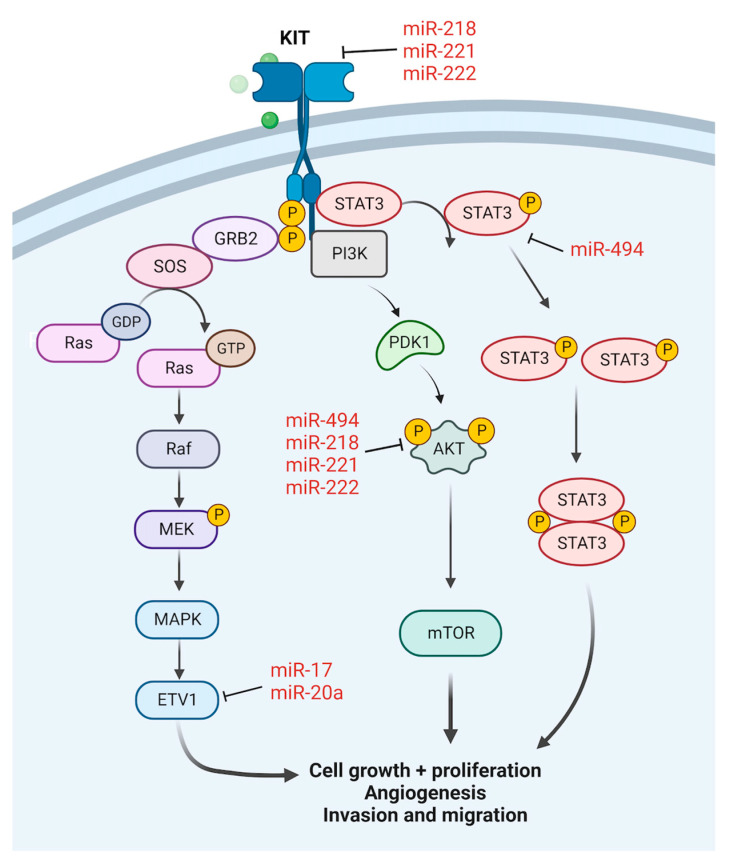
miRNAs regulating KIT and downstream pathways in GIST.

**Figure 3 cancers-15-00577-f003:**
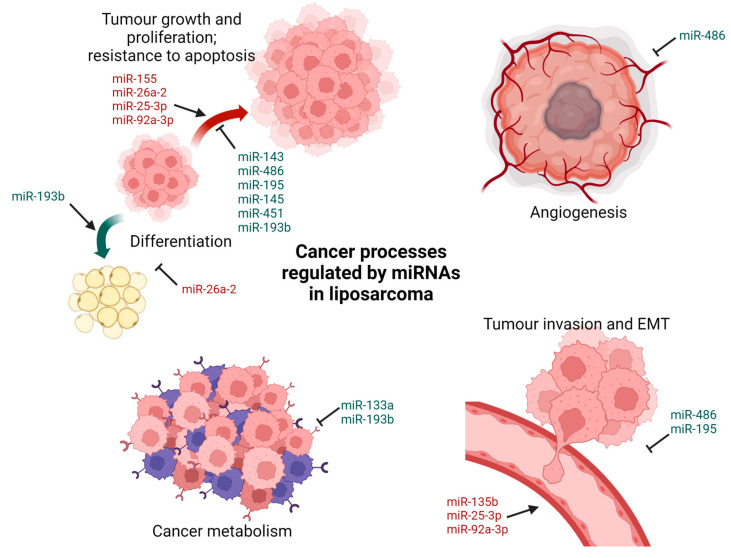
microRNAs involved in the regulation of liposarcoma.

**Figure 4 cancers-15-00577-f004:**
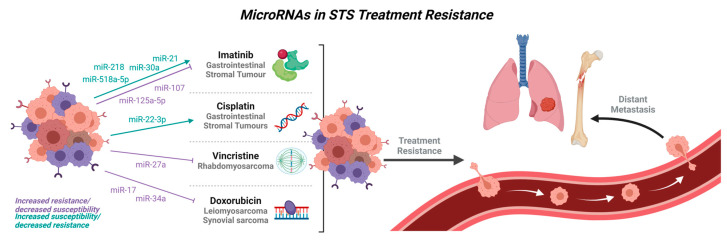
Known microRNAs implicated in treatment resistance of soft tissue sarcomas.

**Table 1 cancers-15-00577-t001:** Differential expression of miRNAs and their roles in cancer development in soft tissue sarcomas.

Soft Tissue Sarcoma	Effect on Cancer Development	microRNA
GIST	Inhibit	miR-494 [23,24]
		miR-218 [25,26,27]
		miR-221/222 [28,29,30]
		miR-17 [30]
		miR-20a [30]
		miR-4510 [31]
		miR-152 [32]
		miR-133b [33]
		miR-518a-5p [34]
		miR-137 [35]
	Promote	miR-374b [36]
		miR-196a [37]
Liposarcoma	Inhibit	miR-143 [38,39]
		miR-486 [40]
		miR-145 [38,41]
		miR-451 [39,41]
		miR-193b [42,43]
		miR-133a [44]
		miR-195 [45]
	Promote	miR-155 [39,46,47,48]
		miR-26a-2 [38,49,50]
		miR-135b [51]
		miR-25-3p [52]
		miR-92a-3p [52]
		miR-3613-3p [53]
Rhabdomyosarcoma	Inhibit	miR-206 [54,55,56,57,58,59,60,61]
		miR-1 [55,56,62,63]
		miR-29 [56,64,65]
		miR-26a [66,67]
		miR-7 [68,69]
		miR-324-5p [69]
		miR-378 family [70]
		miR-133a [62]
		miR-133b [63]
		miR-450b-5p [71]
		miR-203 [72]
		miR-411-5p [73]
		miR-221/222 [74]
		miR-214 [75]
		miR-101 [76]
		miR-874 [77]
		miR-410-3p [78]
	Promote	miR-27a [79,80]
		miR-486-5p [74]
Malignant peripheral nerve sheath tumour	Inhibit	miR-204 [81]
		miR-30d [82]
		miR-30a [83]
		miR-200b [83]
		miR-34a [84]
	Promote	miR-21 [85]
		miR-801 [86]
		miR-214 [86]
Leiomyosarcoma	Inhibit	miR-1246 [87]
		miR-191-5p [87]
		miR-34a [88]
		miR-152 [89]
	Promote	miR-181b [90]
		miR-320a [91]
Synovial sarcoma	Inhibit	miR-494-3p [92]
		miR-126 [93]
	Promote	Let-7e [94]
		miR-99b [94]
		miR-92b-3p [95]
		miR-214 [96]
		miR-9 [97]
		miR-17 [98]
Fibrosarcoma	Inhibit	miR-29 [99]
		miR-197 [100]
	Promote	miR-520c [101]
		miR-373 [101]
Angiosarcoma	Inhibit	miR-497-5p [102]
		miR-210 [103]
		miR-340 [104]
	Promote	-

**Table 2 cancers-15-00577-t002:** miRNAs that prognosticate for poor survival and metastasis in soft tissue sarcoma.

GIST	Poor patient survival	miR-494 (downregulation) [24]
		miR-133b (downregulation) [33]
		miR-1915 (downregulation) [161]
		miR-196a (overexpression) [37]
		let-7e (downregulation) [162]
	Increased metastatic risk	miR-494 (downregulation) [24]
		miR-133b (downregulation) [33]
		miR-1915 (downregulation) [161]
		miR-186 (downregulation) [163]
		miR-196a (overexpression) [37]
		miR-215-5p (downregulation) [164]
Liposarcoma	Poor patient survival	miR-26a-2 (overexpression) [49]
		miR-135b (overexpression) [51]
		miR-155 (overexpression) [39]
	Increased metastatic risk	miR-135b (overexpression) [51]
Rhabdomyosarcoma	Poor patient survival	miR-206 (downregulation) [61] miR-26a (downregulation) [67]
	Increased metastatic risk	miR-206 (downregulation) [61] miR-486-5p (overexpression) [74]
Leiomyosarcoma	Poor patient survival	miR-181b (downregulation) [90]
	Increased metastatic risk	miR-15a (overexpression) [138]
		miR-92a (overexpression) [138]
		miR-31 (downregulation) [138]
Synovial sarcoma	Poor patient survival	miR-214 (overexpression) [96]
	Increased metastatic risk	miR-494-3p (downregulation) [92]

**Table 3 cancers-15-00577-t003:** miRNAs implicated in treatment resistance in soft tissue sarcomas.

Type of Treatment	Soft Tissue Sarcoma	microRNA Involvement
Imatinib	GIST	miR-218 [26]
		miR-518a-5p [34]
		miR-130a [170]
		miR-320a [171]
		miR-21 [168]
		miR-30a [169]
		miR-125a-5p [161,172]
		miR-107 [161]
		miR-518e-5p [173]
Vincristine	Rhabdomyosarcoma	miR-27a [80]
Doxorubicin resistance	Leiomyosarcoma	miR-34a [88]
	Synovial sarcoma	miR-17 [98]

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
