# Peer review of "MicroRNAs in the Pathogenesis, Prognostication and Prediction of Treatment Resistance in Soft Tissue Sarcomas"

_cancers, 2023, doi:10.3390/cancers15030577_

Round 1
Reviewer 1 Report
Overall an excellent manuscript, and excellent review. Well presented and described, with good flow and conclusion. I would accept in its present form.
Author Response
Response to Reviewer 1:
Overall an excellent manuscript, and excellent review. Well presented and described, with good flow and conclusion. I would accept in its present form.
Author response: Thank you for your kind feedback.
Reviewer 2 Report
This represents an excellent, thorough, and well organized review on miRNA in STS.
Author Response
Response to Reviewer 2:
This represents an excellent, thorough, and well organized review on miRNA in STS.
Author response: Thank you for your kind feedback.
Reviewer 3 Report
Teo et al. revised the role of miRNAs in the pathogenesis, prognostication and in treatment resistance of soft tissue sarcomas (STS). They analyzed in detail the implications of miRNAs for the most frequent tumors (GIST, LPS, RMS, LMS and SS). For the treatment resistance, literature evidence is cited for imatinib, vincristine/entinostat and doxorubicin. The authors discussed the potential use and limitations of miRNA regarding the use in treatment of STS.
The manuscript is very well organized including explanatory figures and tables that facilitate the comprehension of this topic to the reader.
Minor correction
In Table 1 there is a full line under “Liposarcoma” that should be shortened in order to include “Inhibit” and “Promote” associated with “Liposarcoma”.
Author Response
Response to Reviewer 3:
Teo et al. revised the role of miRNAs in the pathogenesis, prognostication and in treatment resistance of soft tissue sarcomas (STS). They analyzed in detail the implications of miRNAs for the most frequent tumors (GIST, LPS, RMS, LMS and SS). For the treatment resistance, literature evidence is cited for imatinib, vincristine/entinostat and doxorubicin. The authors discussed the potential use and limitations of miRNA regarding the use in treatment of STS.
The manuscript is very well organized including explanatory figures and tables that facilitate the comprehension of this topic to the reader.
Minor correction
In Table 1 there is a full line under “Liposarcoma” that should be shortened in order to include “Inhibit” and “Promote” associated with “Liposarcoma”.
Author response: Thank you for your kind feedback. We have made the necessary changes to Table 1.